# Predictors and prognosis of population-based subjective cognitive decline: longitudinal evidence from the Caerphilly Prospective Study (CaPS)

Harriet A Ball ,[1] Elizabeth Coulthard,[2] Mark Fish,[3] Antony Bayer,[4] John Gallacher,[5] Yoav Ben-Shlomo [1]

[1]Population Health Sciences, University of Bristol, Bristol, UK
[2]Translational Health Sciences, University of Bristol Medical School, Bristol, UK
[3]Royal Devon University Healthcare NHS Foundation Trust, Exeter, UK
[4]Institute of Primary Care and Public Health, Cardiff University, Cardiff, UK
[5]Department of Psychiatry, University of Oxford, Oxford, UK

**Correspondence to**
Dr Harriet A Ball;
harriet.ball@bristol.ac.uk

## ABSTRACT

**Objectives** To understand associations between the subjective experience of cognitive decline and objective cognition. This subjective experience is often conceptualised as an early step towards neurodegeneration, but this has not been scrutinised at the population level. An alternative explanation is poor meta-cognition, the extreme of which is seen in functional cognitive disorder (FCD).

**Design** Prospective cohort (Caerphilly Prospective Study).

**Setting** Population-based, South Wales, UK.

**Participants** This men-only study began in 1979; 1225 men participated at an average age of 73 in 2002–2004, including assessments of simple subjective cognitive decline (sSCD, defined as a subjective report of worsening memory or concentration). Dementia outcomes were followed up to 2012–2014. Data on non-completers was additionally obtained from death certificates and local health records.

**Primary and secondary outcome measures** The primary outcome measure was incident dementia over 10 years. Secondary outcome measures included prospective change in objective cognition and cross-sectional cognitive internal inconsistency (the existence of a cognitive ability at some times, and its absence at other times, with no intervening explanatory factors except for focus of attention).

**Results** sSCD was common (30%) and only weakly associated with prior objective cognitive decline (sensitivity 36% (95% CI 30 to 42) and specificity 72% (95% CI 68 to 75)). Independent predictors of sSCD were older age, poor sleep quality and higher trait anxiety. Those with sSCD did not have excess cognitive internal inconsistency, but results suggested a mild attentional deficit. sSCD did not predict objective cognitive change (linear regression coefficient −0.01 (95% CI −0.13 to 0.15)) nor dementia (odds ratio 1.35 (0.61 to 2.99)) 10 years later.

**Conclusions** sSCD is weakly associated with prior objective cognitive decline and does not predict future cognition. Prior sleep difficulties and anxiety were the most robust predictors of sSCD. sSCD in the absence of objective decline appears to be a highly prevalent example of poor meta-cognition (ie, poor self-awareness of cognitive performance), which could be a driver for later FCD.

## STRENGTHS AND LIMITATIONS OF THIS STUDY

⇒ The design was prospective and population-based, with a detailed collection of many potential risk or confounding factors, from middle to older age.

⇒ Outcome measures included objective test scores and clinical dementia diagnoses (including local clinic and death certificate records for those who did not participate to the end).

⇒ The study only included men from the outset, so we cannot extrapolate the results to women.

## BACKGROUND

We know that cognitive symptoms are common, and many people who attend memory clinics due to cognitive symptoms turn out not to have a progressive cognitive disorder.[1] The link between subjective reports of cognitive decline (SCD) and objective evidence of cognitive decline is surprisingly weak,[2–4] or sometimes found only in subgroups. An international working group (SCD-I), has published criteria for 'pre-mild cognitive impairment (MCI) SCD' as those who experience persistent decline in cognitive ability relative to their baseline, despite scoring in normal ranges on objective cognitive tests, and excluding those meeting criteria for MCI or dementia, or symptoms that are explained by any medical, psychiatric or neurological disease (except Alzheimer's disease), medication or substance abuse.[5] This definition has been created to harmonise research exploring evidence of neurodegeneration. However, this definition excludes many in the population with a new-onset subjective experience of cognitive decline (which is important since this experience, regardless of the exclusions in the definition, drives their concern and potential presentation to medical services). We define *simple* SCD (sSCD) as purely a subjective

experience, irrespective of objective cognition or medical or psychiatric disorders.

A multi-cohort study found that the risk of dementia in those with SCD (as defined by the SCD-I) was driven very largely by clinic-based rather than community-based or population-based samples.[6] Similarly, a population-level association between SCD and amyloid status was found (only once controlling for anxiety), but dropped out when potential confounders were controlled for.[7] In the English Longitudinal Study of Ageing cohort,[8] although objective cognitive changes predicted subsequent worsening in subjective cognition, deterioration in subjective cognition did not prospectively predict objective cognition. Therefore, isolated subjective cognitive problems *often do not* herald the start of a progressive neurodegenerative process. Research attention is now focusing on a narrower group who have extra clinical features ('SCD-plus') but who are yet further from the common population experience of subjective worsening in cognitive ability. 'SCD-plus' refers to subjective cognitive decline plus at least one additional feature, including: amnestic rather than non-amnestic complaints; onset <5 years ago; onset age more than 60 years; concern associated with SCD; persistence of SCD over time; help-seeking and confirmation of decline by an informant.[9] 'SCD plus' has been associated with a higher risk of developing objective cognitive deficits,[10] or frank dementia, than those with SCD alone.[11] But it is also likely that 'SCD plus' would be overrepresented in clinical populations relative to those in the general population.

The aim of this paper is to explore population-based associations with the simple subjective experience of cognitive decline (sSCD), since outcomes are likely to be different than when examining clinic-based populations that are less representative. The latter are likely to have worse outcomes, which might make more aggressive investigation and potential treatment appropriate, but that may not hold for the general population with subjective cognitive problems. This is important to consider since the typical cases seen currently in clinics may expand and include different types of people in the light of novel therapies and wider public awareness of neurodegenerative problems. This paper also takes a more nuanced and dimensional view of any association with objective cognitive change, noting individual differences in test scores rather than whether someone is just above versus just below a cut-off for 'abnormal'.

A subjective experience of cognitive difficulty that outstrips objective evidence, when severe and impairing, is a key feature of functional cognitive disorder (FCD)[12]; but a milder experience might be common in the wider population. Functional symptoms (which can wax and wane) therefore might explain *some* of the 'recovery' of people with scores in the MCI range back up to the normal range (up to one-third in community-based samples).[13] Another key feature of FCD is cognitive 'internal inconsistency', which is when someone struggles to perform a cognitive ability for which they are demonstrably capable

without any external factor intervening, aside from perhaps a change in focus of attention.[12 14]

Population-based analyses show subjective cognitive concerns are more common in people who are older, of lower social class and less educated[15] and are associated with worse quality of life and higher anxiety.[16] Sleep disturbance is associated with objective cognitive dysfunction, more so at older ages,[17] but less has been reported about its association with subjective cognition.

The current paper uses a prospective cohort of men in Wales, with in-depth cognitive assessments and data spanning over three decades, to examine the prevalence of sSCD, prospective associations from potential causal factors, and objective cognitive change. Our research questions were as follows:

1. Is sSCD a good indicator of prior objective cognitive decline? This is relevant because clinicians might suspect SCD to reflect some objective change, perhaps too subtle to be detected on simple testing with broad population normal values.
2. Do those with sSCD, in the absence of objective decline, also have features of cognitive internal inconsistency?
3. Does sSCD have different risk factors compared with those for objective cognitive decline?
4. Is sSCD associated with a risk of future objective cognitive decline or dementia?

## METHODS

The Caerphilly Prospective Study (CaPS) began in 1979 (online supplemental figure 1).[18–20] Men aged 45–59 were identified from electoral rolls and general practice lists in Caerphilly and surrounding villages in South Wales, UK. Women were excluded as the initial focus was on cardiovascular health and it was felt that too few events would be accrued in the early follow-up. Beginning in phase 3 (age 55–69 years), cognitive assessments were added, including the Cambridge Cognitive Examination (CAMCOG),[21] the Alice Heim 4 (AH4) test[22] and the Rivermead Behavioural Memory Test (RBMT,[23] which includes immediate recall of a 21-clause story and delayed recall after 20 min). Cognitive assessments were also repeated at phases 4 and 5. At phase 5 (mean age 73 years), a subset of participants (those who could not complete the CAMCOG, scored 82 or fewer, or showed at least a 10-point decline over time) were invited to a clinical assessment to identify those who had dementia and those who had 'cognitive impairment not dementia' (CIND, which includes those with MCI). Informants (eg, spouses or children) also rated the change in participants' day-to-day abilities (Informant Questionnaire on Cognitive Decline in the Elderly (IQCODE)).[24] A further assessment of *likely* dementia status was made holistically at phase 7 by a cognitive disorders clinician (AB), using all available information, including the longitudinal CAMCOG score, IQCODE and all medical records. Those who had previously been classified as having dementia

at phase 5 were assumed to still have this diagnosis and were not re-evaluated. By this phase (mean age 82 years), some men had other health problems, making it difficult to complete the cognitive assessments, leading to altered scores related to factors such as hearing, vision or manual dexterity.

We have followed the Strengthening the Reporting of Observational Studies in Epidemiology cohort reporting guidelines.[25]

In assessing the decline in objective cognitive test scores from phases 3 to 5, we focused first on an instrument typically used as a dementia screen (CAMCOG) and second a scale more often used to indicate reasoning skills in the general population (AH4). The mean score at phases 3 and 4 (or just one if the other was missing) was subtracted from the score at phase 5 and divided by the time interval to indicate the rate of change over time (years). We defined 'objective cognitive decline' as a decline of one SD below the mean decline of those men who had neither dementia nor CIND at phase 5. Although an arbitrary cut-off, this value was chosen to capture a potentially subtle difference from an expected average decline over 8–12 years, which is unlikely to be due to merely random error; with the benefit of premorbid detailed cognitive assessment, this difference might be smaller than what is felt to be a meaningful decline in a clinical setting.

At phase 5, men were asked about their subjective cognition via two questions: In the last 12 months, my *memory/concentration* (both asked independently) has improved, not changed, got a little worse and got a lot worse. These we recoded as 0=improved/no change, 1=little worse and 2=lot worse to produce scores (0–2) for memory and concentration. Scores on these two items were correlated (Spearman's rho 0.64, p<0.001) and we combined them to generate a total score (0–4). We also created a binary variable (sSCD) defined as a score of 2 or more out of 4. Therefore, our definition of SCD is more basic than SCD as defined earlier by the international working group.

'Cognitive internal inconsistency' was not the focus of this study at the time of data collection. Performance validity tests exist but are protracted and were not performed as part of this study; further, the literature around these tests mainly focus on identification of conscious malingering, for example, the Test Of Memory Malingering.[26] We therefore looked for ways to identify unambiguous cognitive internal inconsistencies in routinely collected data. We pragmatically operationalised this as delayed recall being paradoxically higher than immediate recall on the RBMT story recall task (ie, looking at a memory component of internal inconsistency) and compared the frequency of this across those with and without sSCD. This is very far from a definitive test for FCD, but we would expect it to be present more often if a group included an excess of people with FCD. Due to the constraints of the data collected, it was not possible to identify a similarly unambiguous signal of

internal inconsistency for cognitive components other than memory.

We considered the following variables as important confounders: age, social class (manual vs non-manual), years of education (stopping before or after age 14 years) and premorbid IQ proxied by the National Adult Reading Test (NART).[27] We assessed neurodegenerative risk markers at phases 2–4. These included probable ischaemic heart disease (assessed through hospital records and ECG changes), body mass index (BMI) and waist:hip ratio, smoking status and medication history indicating vascular risk (ie, medications whose indications were unequivocally for blood pressure, vascular disease, diabetes or lipid lowering). Alcohol exposure was indexed both as a binary marker of avoidance ('teetotal' at *any* of phases 2–4) and as a binary indicator of the highest cumulative alcohol use at phases 2–4 (those drinking an average of four or more units per day across the period), due to previous work suggesting a non-linear association between alcohol and cognition.[28] At phase 3, we also assessed mood symptoms using the chronic scoring of the General Health Questionnaire (GHQ), the Spielberger State-Trait Anxiety Inventory and the use of any antidepressant medication. Sleep problems were assessed using items from the Wisconsin Sleep Questionnaire, divided into 'experiencing poor sleep' and 'breathing-disordered sleep' (see online supplemental material).

## Statistical analysis plan

The present study includes men who took part in phase 5, including the cognitive assessments (n=1225). We calculated the prevalence of sSCD at phase 5 as the number of cases divided by the population at risk. We compared objective cognitive decline between phases 3 and 5, either as a continuous score (t-test) or binary measure 'objective cognitive decline' ($\chi^2$ test, using CAMCOG and AH4 separately) for those with and without sSCD at phase 5. We also derived the sensitivity and specificity of sSCD in relation to objective decline (Yes/No) as the gold standard. Among those with no objective cognitive decline by phase 5, we tested for an association between sSCD and cognitive internal inconsistency using a $\chi^2$ test. We ran multivariable logistic regression models to identify possible causal contributors (measured at phases 2–4) to sSCD (at phase 5). We tested whether sSCD (at phase 5) predicted either a decline in CAMCOG (linear regression) or a worsening cognitive severity category (logistic regression) between phases 5 and 7.

To address the potential bias when undertaking a complete case analysis of potential causal contributors to sSCD, we included multiple imputation with chained equations (MICE) and pooled estimates using Rubin's rules across 50 imputed datasets. The imputation model used all the variables included in the analysis model (ie, all the potential predictors of sSCD as well as sSCD itself). To address potential survivor bias, we included a MICE

**Table 1** Relationship between change in objective cognition performance (phases 3–5) and self-reported simple subjective cognitive decline (phase 5)

| Decline in cognitive test score (annual rate) | Mean (SD) | | Difference (95% CI) | P value | Adjusted R-squared* |
|---|---|---|---|---|---|
| | No subjective cognitive decline | Subjective cognitive decline | | | |
| Whole sample | | | | | |
| CAMCOG | 0.17 (0.63) | 0.35 (0.91) | 0.18 (0.8 to 0.27) | <0.001 | 0.01 |
| AH4 | 0.08 (0.61) | 0.26 (0.66) | 0.18 (0.09 to 0.27) | <0.001 | 0.02 |
| Excluding those with dementia at phase 5 | | | | | |
| CAMCOG | 0.13 (0.55) | 0.17 (0.49) | 0.04 (−0.04 to 0.11) | 0.36 | <0.001 |
| AH4 | 0.07 (0.59) | 0.22 (0.62) | 0.16 (0.07 to 0.24) | <0.001 | 0.01 |

*The proportion of variance in the dependent variable that can be predicted by the independent variable
AH4, Alice Heim 4; CAMCOG, Cambridge Cognitive Examination.

model when looking at phase 7 outcomes, using the same principles and the same imputed variables as in the aforementioned imputation, except we additionally included all variables that were in the phase 7 analysis model in the imputation model.

As sensitivity analyses, we repeated the above excluding those with a pre-existing diagnosis of dementia. This is because in a clinical setting, it is usually relatively straightforward to identify people whose cognitive impairment is severe enough to be categorised as dementia; the diagnostic difficulty lies in understanding causes among those whose cognitive difficulty is milder. In addition, it excludes those who might have missing data on items about subjective cognition due to being severely cognitively impaired (which could introduce bias). We also repeated analyses looking at the memory-only versus

**Table 2** The diagnostic utility of sSCD for objective cognitive performance

| | Objective cognitive decline,* n (%) | No objective cognitive decline, n (%) | P value | Sensitivity (95% CI) | Specificity (95% CI) |
|---|---|---|---|---|---|
| Whole sample | | | | | |
| | CAMCOG+ve | CAMCOG−ve | | | |
| sSCD | 90 (9) | 201 (21) | 0.03 | 36 (30 to 42) | 72 (68 to 75) |
| No sSCD | 163 (17) | 515 (53) | | | |
| | AH4+ve | AH4−ve | | | |
| sSCD | 71 (8) | 205 (22) | <0.001 | 44 (36 to 52) | 73 (70 to 76) |
| No sSCD | 91 (10) | 558 (60) | | | |
| | CAMCOG or AH4 +ve | CAMCOG or AH4 −ve | | | |
| sSCD | 132 (14) | 159 (16) | <0.001 | 38 (33 to 43) | 74 (71 to 78) |
| NO sSCD | 216 (22) | 462 (48) | | | |
| Excluding dementia | | | | | |
| | CAMCOG+ve | CAMCOG−ve | | | |
| sSCD | 67 (7) | 199 (21) | 0.36 | 31 (25 to 38) | 72 (69 to 75) |
| No sSCD | 148 (16) | 513 (55) | | | |
| | AH4+ve | AH4−ve | | | |
| sSCD | 61 (7) | 196 (22) | <0.001 | 42 (34 to 51) | 74 (71 to 77) |
| NO sSCD | 83 (9) | 552 (62) | | | |
| | CAMCOG or AH4 +ve | CAMCOG or AH4 −ve | | | |
| sSCD | 109 (12) | 157 (17) | 0.002 | 35 (30 to 41) | 75 (71 to 78) |
| No sSCD | 201 (22) | 460 (50) | | | |

*Continuous scoring converted to a binary variable, +ve/−ve indicates whether a participant had declined more/less than one SD from the mean rate, calculated among men without CIND or dementia, between phases 3 and 5.
AH4, Alice Heim 4; CAMCOG, Cambridge Cognitive Examination; CIND, cognitive impairment not dementia; sSCD, simple subjective cognitive decline.

**Table 3** sSCD and cognitive internal inconsistency, among people without objective cognitive decline (at phase 5), n=619.

| Men with no objective cognitive decline, phase 5 | Subjective report of cognition | | |
|---|---|---|---|
| Rivermead story recall task | No sSCD, mean (SD) | sSCD, mean (SD) | P value |
| IR | 6.15 (2.82) | 5.55 (2.59) | 0.02 |
| DR | 5.29 (2.78) | 4.73 (2.59) | 0.03 |
| IR–DR | 0.84 (1.45) | 0.83 (1.41) | 0.92 |
| Retention: DR/IR | 0.86 (0.37) | 0.86 (0.35) | 0.58 |
| Relative difference: (IR–DR)/IR | 0.12 (0.37) | 0.14 (0.35) | 0.58 |
| Consistency of recall ability | n (%) | | $\chi^2$ P value |
| 'Consistent' (DR≤IR) | 370 (80.8) | 130 (82.3) | 0.68 |
| 'Inconsistent' (DR>IR) | 88 (19.2) | 28 (17.7) | |

DR, delayed recall; IR, immediate recall; sSCD, simple subjective cognitive decline.

concentration-only aspects of subjectively reported cognition. We performed a post hoc power calculation, though this is controversial and may be misleading.[29]

Analyses were conducted in STATA v16.

## Patient and public involvement

This study began in the 1970s, when it was not normal practice to involve participants in the design and co-production of research. However, the researchers kept the participants well informed of the research progress and key findings through regular newsletters, including a 25th follow-up birthday party where survivors met the research team and the Welsh Minister of Health. Over the study duration, ad hoc feedback has occurred, with study participants writing or speaking to the research team with their views about the study. For example, a sensitive question was dropped based on participants' comments.

## RESULTS

### Sociodemographics and frequency of risk variables

One thousand two hundred and twenty-five men took part in phase 5, including the completion of cognitive assessments, with a mean age of 73 (range 65–83) years (online supplemental figure 1). Sixty-two per cent of men (n=720) were of the manual social class. Forty-six per cent (n=507) had left school by age 14 (in later analyses, we used NART-IQ as a more normally distributed proxy for the amount of education received). Seventy-five (6.1%) were assessed as having dementia, and 192 (15.7%) as having CIND. Three hundred and twenty-six participants (30.6%) had sSCD. Unadjusted, subjects with sSCD were more likely to have CIND, dementia, lower education, probable ischaemic heart disease, be older, report worse sleep, have a worse GHQ score and show slightly greater declines in their CAMCOG and AH4 scores (online supplemental table 1). sSCD status was missing for 160/1225 (13%) among the whole cohort, 42/192 (22%) among those with CIND and 25/75 (33%) among those with dementia. Only 15 men (1.2%) were

on antidepressant medication; due to low prevalence, this variable was not analysed further.

Of those who did not already have dementia at phase 5, 37/573 (6.5%) had possible dementia by phase 7. Of those who did not already have CIND at phase 5, 84/528 (15.9%) had CIND or dementia by phase 7. Of those who were classified as not having dementia at phase 5, 575/1148 (50.1%) had missing data on this measure at phase 7.

### Question 1: is subjective cognitive decline (at phase 5) a good indicator of prior objective cognitive decline?

Objective CAMCOG cognitive change was scored continuously, and the average drop (from the baseline of phase 3 or 4 to phase 5) was 2.7 points per year in those developing dementia, 0.7 per year in those developing CIND and 0.06 per year in those without either by phase 5. The equivalent figures for AH4 were 0.9, 0.2 and 0.1, respectively. Objective changes in cognition were slightly worse in those with sSCD (table 1), but the proportion of variance explained was very low. These associations were attenuated when excluding those with dementia, and this (unadjusted) association remained statistically significant for AH4 (but not for CAMCOG).

Of people who reported sSCD, the majority did not have objective cognitive decline as defined by ≥1 SD decline (table 2). The utility of sSCD as a screening tool for objective decline was poor, with a sensitivity of 36–44% and a specificity of 72–75%.

### Question 2: do those with subjective cognitive decline, but no objective decline (at phase 5), have features of cognitive internal inconsistency?

In the Rivermead story recall test, subjects reporting sSCD did slightly worse than others on immediate recall (5.55 vs 6.15, p=0.02), and then proportionately worse on delayed recall (4.73 vs 5.29, p=0.03). The worse score on delayed recall was in proportion to the worse performance on immediate recall and more reflective of a mild attentional deficit than a true amnestic difficulty. There

**Table 4** Possible risk factors (phases 2–4) for simple subjective cognitive decline (phase 5)

| | Adjusted OR (95% CI) (n=601) | Adjusted OR (95% CI) excluding dementia (n=580) |
|---|---|---|
| Age (years) | 1.12 (1.06 to 1.17) | 1.11 (1.05 to 1.17) |
| Social class (manual/non-manual) | 0.99 (0.65 to 1.51) | 0.95 (0.62 to 1.46) |
| Premorbid IQ (x 0.1) | 1.06 (0.84 to 1.33) | 1.05 (0.83 to 1.32) |
| High alcohol use (binary) | 1.07 (0.63 to 1.83) | 1.20 (0.70 to 2.07) |
| Teetotal (binary) | 0.64 (0.31 to 1.32) | 0.60 (0.29 to 1.27) |
| Probable ischaemic heart disease (binary) | 0.98 (0.59 to 1.62) | 1.03 (0.61 to 1.72) |
| Number of vascular medications | 1.18 (0.92 to 1.50) | 1.20 (0.93 to 1.54) |
| Smoking (binary) | 0.89 (0.60 to 1.34) | 0.85 (0.56 to 1.30) |
| Waist:hip ratio (×10) | 0.93 (0.66 to 1.31) | 0.89 (0.63 to 1.27) |
| Poor sleep | 1.52 (1.19 to 1.94) | 1.55 (1.21 to 1.99) |
| Sleep-disordered breathing | 1.13 (0.91 to 1.40) | 1.12 (0.90 to 1.40) |
| Mood symptoms (GHQ score: chronic)* | 1.27 (1.01 to 1.59) | 1.23 (0.97 to 1.55) |
| Trait anxiety (STAI)* | 1.38 (1.09 to 1.74) | 1.39 (1.09 to 1.76) |
| Rate of decline CAMCOG | 1.18 (0.83 to 1.69) | 1.18 (0.72 to 1.95) |
| Rate of decline AH4 | 1.33 (0.95 to 1.85) | 1.36 (0.97 to 1.92) |

*Standardised

AH4, Alice Heim 4; CAMCOG, Cambridge Cognitive Examination; GHQ, General Health Questionnaire; STAI, State-Trait Anxiety Inventory.

was no evidence of excess cognitive internal inconsistency (indicated by a higher score on delayed than immediate recall) when comparing those with and without sSCD (0.83 vs 0.84, p=0.92) (table 3 and online supplemental figure 2).

### Question 3: which characteristics predict subjective cognitive decline (at phase 5), and are these different to risk factors traditionally linked to objective cognitive decline?

Variables that predicted sSCD in unadjusted logistic models included: age, probable ischaemic heart disease, vascular medications, experiencing poor sleep, breathing-disordered sleep, mood symptoms, trait anxiety, rate of decline in CAMCOG and rate of decline in AH4 (table 4, and online supplemental table 3). There was no unadjusted association with social class, premorbid IQ, alcohol use (either high use or teetotal status), tobacco exposure, BMI, waist:hip ratio or baseline measures of CAMCOG and AH4. However, in a model simultaneously adjusting for all these factors, the only significant independent predictors remaining were age, experiencing poor sleep, mood symptoms and trait anxiety (respective ORs: 1.12, 1.52, 1.27 and 1.38). These associations (except mood symptoms, which became borderline non-significant) remained when excluding participants with a clinical diagnosis of dementia at phase 5.

Due to missing data, the adjusted complete case analysis only included 601 participants. This model, but with multiple imputation (n=1225), found the same significant independent predictors with the addition of one variable (rate of decline in AH4, OR 1.50, 1.16 to 1.95, see online supplemental table 3).

### Question 4: does subjective cognitive decline (phase 5) predict future CIND and dementia (phase 7)?

sSCD at phase 5 did not predict worsening cognition at phase 7 (table 5), when examined either by diagnosis (new CIND or new dementia) or by change in CAMCOG score. Due to attrition to phase 7, this was assessed both by complete case analysis and multiple imputation.

### Sensitivity analyses and post hoc power calculation

The results of analyses excluding those with a diagnosis of dementia are reported above.

We replaced overall sSCD with reported complaints in either memory-only or concentration-only. These showed the same patterns of association with measures of objective cognition, internal inconsistency (lack of association), possible confounders and risk factors, and future dementia (lack of association).

In testing the hypothesis that sSCD may or may not predict the combined outcome of incident CIND and dementia, we ran several power calculations under the assumption of 5%, 7.5% and 10% increased risk for the sSCD group over the global risk. The ratio of sSCD to non-sSCD was 0.44 (at phase 5), and the proportion of people who newly developed CIND or dementia cases (between phases 5 and 7) was 0.21. The power for these two-sample comparisons, using a two-sided alpha of 0.05, was, respectively, 0.54, 0.84 and 0.99.

### DISCUSSION

In this population-based sample of 1225 older men in South Wales, we found only a weak association between

**Table 5** sSCD does not predict future cognitive outcomes (new CIND or dementia)

| Predictor at phase 5: sSCD | | | |
|---|---|---|---|
| Outcome at phase 7 | CIND or dementia | Dementia | Decline in CAMCOG between phases 5 and 7* |
| Participants | Excluding those with CIND or dementia at phase 5 | Excluding those with dementia at phase 5 | Excluding those with dementia at phase 5 |
| Model | OR (95% CI) | OR (95% CI) | Linear coefficient (95% CI) |
| Complete case analysis | 0.92 (0.49 to 1.73) | 1.35 (0.61 to 2.99) | 0.01 (−0.13 to 0.15) |
| Multiple imputation | 0.98 (0.56 to 1.74) | 1.18 (0.53 to 2.61) | −0.02 (−0.19 to 0.15) |

All models controlled for: age, CAMCOG score at phase 5.
*The CAMCOG analysis also controlled for a dummy variable indicating CAMCOG points lost for hearing, vision or manual dexterity at phase 7.
CAMCOG, Cambridge Cognitive Examination; CIND, cognitive impairment not dementia; sSCD, simple subjective cognitive decline.

objective cognitive change and subjective reports of that same change, partially accounted for by some of the subjective group having dementia. Those who had subjective cognitive concerns without objective cognitive decline, nonetheless, showed a mild deficit in attentional processes. At the population level, sSCD has low sensitivity and specificity to identify objective evidence of cognitive decline (both missing many people who lack insight into their cognitive problems and detecting people who are concerned in the absence of objective problems). Independent predictors for sSCD included preceding anxiety or mood symptoms, and poor self-reported sleep, as well as older age, but not other variables known to be risk factors for neurodegeneration. sSCD, adjusted for objective cognition, did not predict later CIND or dementia.

This is the first large population-based cohort study of subjective experience of cognitive decline that can look at both predictors and future clinical outcomes, with a wide range of potential confounders measured at younger ages, and hence is less prone to bias and reverse causation. We found no prospective association between sSCD and later objective cognitive decline (either in the numerical score or severity category, ie, CIND or dementia). Naturally, a larger sample size would have given greater power to detect a more modest effect. However, this negative finding suggests that subjective cognitive symptoms, by themselves, are not substantially (on a population level) the manifestation of a neurodegenerative condition. We found a weak association between prior objective cognitive decline and subjective reports of decline in cognition. The interpretation of this association is complex since those with severely impaired cognition might lack insight or even not understand the question, but there are also a large number of people in the general population who feel their cognition is suboptimal. Previous studies that have found a more convincing association have largely been clinic-based. The reasons for this discrepancy likely include differences in populations according to who seeks and can access clinical assessment (which could include socioeconomic differences, age, comorbidities and ethnicity).[30] Most clinic-based studies of SCD often have participants with 'SCD-plus' (ie, extra factors, such as partner concern, that make neurodegeneration more likely and may increase their chances of reaching clinic). The choice of test and cut-off and the cross-sectional versus decline measure can also affect results. Our decline measure is potentially a quite sensitive measure of change over time, given that we had a premorbid baseline measure, which is absent in clinic-based studies that begin when patients attend. The exact question framing also matters regarding self-reported cognitive difficulties, since some items are more likely to index normal ageing, whereas others index pathological changes.[31–33]

Of our two objective cognitive tests, AH4 showed a slightly more robust association with sSCD than did CAMCOG (AH4 assesses complex verbal and numerical reasoning, analogous to an IQ test[22]; CAMCOG measures faculties more commonly assessed in the dementia clinic[21]). Perhaps, the population variance in complex reasoning more closely mirrors people's understanding of what it is to have memory or concentration problems. Alternatively, perhaps AH4 captures more meaningful variance in the normal range of cognition, whereas CAMCOG is better at detecting the more severe end (where those with cognitive difficulties often lack insight). Either way, it may be useful to incorporate measures similar to AH4 into neuropsychometry batteries for patients with suspected functional neurological symptoms, as this could represent a measure capable of tracking progress and providing a useful discussion point with patients.

Among men with subjective cognitive symptoms despite a lack of objective cognitive decline, we found they fared slightly worse (compared with those without cognitive symptoms) in a task of immediate recall but no worse than expected on delayed recall of the initially remembered items. This reinforces the suggestion that SCD is linked to a slight deficit in attentional processes (despite not having dipped on overall cognitive test scores over many years). This pattern is distinct from the amnestic processes typically seen in early Alzheimer's disease[34 35]

or the attentional problems alongside wider progressive cognitive problems seen in dementia with Lewy bodies.[36] It is also of note that this group did not exhibit cognitive internal inconsistency (no evidence of scoring *better* on delayed recall than on immediate recall). This is not entirely surprising, as our group is quite distinct from clinic-based patients with FCD in whom this characteristic is a key part of diagnosis[12] (and in whom more than a one-off example of evidence of internal inconsistency should be sought). Attentional problems have been identified in other conditions within the umbrella of functional somatic syndromes when such patients have been tested cognitively.[37] It is also important to recognise that our pragmatic measure of 'cognitive internal inconsistency' is identifiable in approximately one-fifth of normal healthy people (online supplemental figure 2), a surprisingly high number if only taking this one snapshot measurement into account.

We found that sSCD was more common at older ages. The same has been found in individual studies where a broad age range is considered.[8] When comparing across studies of people of differing ages, this association is less robust,[1] probably in part due to heterogeneity between included studies. Reasons for this association could include differences in meta-cognitive processes by age, older people resigning to using workarounds, age-related cultural expectations or the way questions are framed leading to self-rating against different comparator groups, as well as some loss of insight when objective cognitive deficits take over.[38] The dip in subjective cognition with age may reflect the recognised small dip in objective cognitive abilities at older ages, as well as the higher prevalence of neurodegeneration. However, factors typically associated with neurodegenerative change (vascular health, alcohol use) did not predict later sSCD in the fully adjusted model. This was also held in an analysis excluding people with dementia. We interpret this to indicate that SCD is associated with objective cognitive decline and age, but also has important predictors independent of objective cognition. These predictors included prior reports of poor sleep quality and prior anxiety symptoms. This supports the proposition that anxiety and poor sleep may be contributing to sSCD (rather than being a by-product of worry about cognitive change). These risk factors for sSCD may also be related to functional cognitive symptoms. It is also notable that anxiety and poor sleep were relevant in our study, which consisted of older males (whereas the literature and clinical diagnoses of functional disorders focus more on younger females). The association between subjective and objective cognition may be different in women, in particular since women may have an extra mid-life peak of subjective cognitive concern.[15]

Subjective reports of poor sleep often go alongside sSCD; however, when using actigraphy (an objective assessment of sleep quality), the association is less or even reversed.[1] There are likely qualitatively different sleep problems among those with sSCD compared with those with organic changes to sleep in early dementia.

## Clinical implications

Subjectively experiencing cognitive decline in the absence of an objective decline is a marker of poor meta-cognition (ie, poor ability to reflect on and monitor one's own cognitive abilities). This is a process postulated to be important in the development of FCD.[39] The current study demonstrates the high prevalence of subjective cognitive decline in the absence of objective cognitive decline. Subjective cognitive decline is not an independent predictor of later neurodegeneration. Since it is associated with increased self-focused attention,[40] subjective cognitive decline may be a vulnerability point from which some people go on to develop FCD, so clinicians should be aware of this condition and aim to positively diagnose it when this would be beneficial to patient management.[12] Of course, there are many alternative causes of cognitive change (aside from FCD and neurodegeneration), so holistic clinical assessment remains crucial for those with problematic symptoms.

Although low-mood symptoms and reports of poor sleep, which often go alongside low mood, were found to predict subjective cognitive decline, this does *not* simply imply that depression causes subjective cognitive decline. Most of the people in our sample did not have symptoms sufficient to qualify for a clinical diagnosis of depression; rather, these are correlations between milder level symptoms. While depressive symptoms correlate primarily with subjective measures of cognition, they may[41] or may not[8] lead to objective cognitive deficits. This suggests that processes interfering with meta-cognition may affect those with depression, as well as many others without depression.

Subjective cognitive decline is quite a poor indicator of recent objective cognitive decline and does not independently predict future objective decline. These findings are relevant to the current design of memory clinics and potential changes if we enter an era of disease-modifying therapies for neurodegenerative conditions. These services would be overwhelmed if opened up to any and all people with subjective cognitive decline, with little need or benefit from clinical input (and potential iatrogenic harm via heightened concern while awaiting clinical assessment). It is therefore important to capture objective evidence of decline and separate features of clinical concern to identify those more likely to benefit from specialist input.

## Limitations

These data are observational, so we cannot draw strong causal inferences. CAMCOG represents a summation of different cognitive faculties; it would be helpful to be able to examine individual cognitive domains more carefully to better understand which most closely track subjective cognitive decline, and similarly, the same cognitive faculty under different levels of attentional focus (eg, free recall vs recognition vs implicit memory). The nature of subjective cognitive decline could also be probed in more detail, such as the level of concern associated with it. This

study examined older white men only in one region of Wales. We should be cautious about generalising our results to women, ethnic minorities and other geographical populations, especially those from low-income or middle-income countries. Older white men are not the demographic associated with the highest prevalence of functional disorders reported from clinics. Nevertheless, ours is an important group to study, as we empirically don't know how common functional cognitive symptoms may be in the general population, especially since diagnostic labels may be swayed by the higher a priori likelihood of neurodegeneration at older ages.

Some results may be influenced by survival or loss-to-follow-up bias: only men who took part at phase 5 (average age 73) are included; their cognitive trajectories may not be representative of men who died, were lost to follow-up or were too impaired to take part (although examination of death certificates and local health records were used to identify dementia cases among non-attenders). Nonetheless, this study gives greater insight than would a hypothetical study that only began enrolling men at an older age (and thus never included those men from the start) and had no data on risk factors in earlier life. We tried to address loss-to-follow-up at the final phase using multiple imputation; however, these models may still be biased if this is informative censoring, whereby the observed outcome itself predicts missingness.[42] A post hoc power calculation showed reasonable power to detect moderate to large differences between participants with and without subjective cognitive decline. But the sample size and incidence of new CIND and dementia meant we could not exclude a modestly increased risk.

## CONCLUSIONS

Subjective cognitive decline is modestly associated with objective cognitive decline, but these processes appear to have largely distinct underlying mechanisms. Our data provide reassuring evidence that subjects with subjective, but without objective cognitive impairments, are not at increased risk of later CIND or dementia.

**Contributors** MF, AB, JG and YB-S contributed to the conception and design of the study. JG led the data collection. MF and AB established the clinical diagnoses. HAB, YB-S and EC planned the analyses. HAB analysed the data and wrote the first draft. All authors read and approved the final manuscript. HAB is responsible for the overall content as guarantor.

**Funding** HAB is a National Institute for Health Research (NIHR) Academic Clinical Lecturer. YB-S is partially supported by the NIHR Applied Research Collaboration West (NIHR ARC West). JG is partially supported by MRC grant MR/T0333771. CaPS was funded by the Medical Research Council (grant G9824960-E01/1), and phase 5 was funded by a grant from the Alzheimer's Society.

**Disclaimer** The views expressed in this article are those of the author(s) and not necessarily those of the NIHR or the Department of Health and Social Care.

**Competing interests** None declared.

**Patient and public involvement** Patients and/or the public were involved in the design, or conduct, or reporting, or dissemination plans of this research. Refer to the Methods section for further details.

**Patient consent for publication** Not applicable.

**Ethics approval** This study involves human participants and was approved by the Gwent Research Ethics Committee (01/69). Participants gave informed consent to participate in the study before taking part.

**Provenance and peer review** Not commissioned; externally peer reviewed.

**Data availability statement** Data are available upon reasonable request. The datasets analysed during the current study are not publicly available due to the sensitive information contained and the nature of the consent given by the participants. The data that support the findings are available on reasonable request.

**ORCID iDs**
Harriet A Ball http://orcid.org/0000-0002-2137-7582
Yoav Ben-Shlomo http://orcid.org/0000-0001-6648-3007

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
