## [Reviewer comments · BMJ Open]

ARTICLE DETAILS

TITLE (PROVISIONAL)	Predictors and prognosis of population-based subjective cognitive decline: Longitudinal evidence from the Caerphilly Prospective Study (CaPS)
AUTHORS	Ball, Harriet; Coulthard, Liz; Fish, Mark; Bayer, Antony; Gallacher, John; Ben-Shlomo, Yoav

VERSION 1 – REVIEW

REVIEWER	Stephen C L Lau Washington University School of Medicine in Saint Louis
REVIEW RETURNED	27-Apr-2023

GENERAL COMMENTS	Thank you for giving me the opportunity to review this manuscript. This manuscript has some interesting findings and may benefit the readers of BMJ Open. My comments are listed, and I hope that the authors will be willing to address them. Introduction: • Since this study only included men, it will be helpful to include a discussion of potential gender differences from the literature, and included the justification of why only men, but not women, were examined.• Line 77-78: it is unclear why depression was suddenly brought up in the discussion without prior context. It would be helpful to provide the relationship of depression to this study.• It is a bit unclear why SCD-plus was mentioned in the article since it is not the definition used for this study. The author could consider adding a summary of what message they were trying to bring in a sentence that follows the discussion.• It would be better if the hypotheses of the study followed the aims of the study. This will strengthen the coherence of the introduction and help the reader to interpret the two. Methods: • It would be helpful if authors could describe the tested components of the CAMCOG. Since the focus of this study is to examine the association between subjective and objective cognition, it is important to understand whether the cognitive components examined in the assessment approaches are similar.• A section of power analysis for the study is recommended. Results: • It would be helpful for interpretation if the authors could include the numerical analysis results (e.g., coefficient and p-value) to some of the important findings discussed in this section. Discussion: • Since a strength of this study is the longitudinal design; it would be better if the authors can leverage it further to discuss causality with reference to the observed associations.
---

	 • The clinical implication section should be strengthened. Discussion on actionable intervention practice and assessment protocol in relation to the study findings are suggested. • Immediate recall and attentional processes are subcomponents of executive functioning. What is the possible relevance of executive functioning deficit to the findings of this study? • After reading the discussion, I think adding more clarification about cognitive inconsistency will help. Looking at Table 3, inconsistency was defined as DR>IR. Justifications behind this formula are recommended in the main text. And it is also recommended to discuss why was cognitive inconsistency restricted to only memory components.
--	---

REVIEWER	Karen Sverdrup Nasjonalt senter for aldring og helse
REVIEW RETURNED	22-Jun-2023

GENERAL COMMENTS	Thank you for an interesting read, well-structured paper, and well conducted study. I have some minor comments, please see in the following, numbered by the questions in the reviewer's form. 1. Is the research question or study objective clearly defined? Comments: The manuscript could benefit from coherent language in aims (line [I.] 91-92), objectives (l. 111-113?), hypothesis (l. 114-118), statistics (l. 180-188, what RQs are being answered) and subheading of results (l. 206, 218, 235-236, 245, 256). Could the objectives be reframed to research questions (RQs) in line with results subheadings? E.g. intro: "...explore population-based associations with, and predictions from, the subjective experience of cognitive decline (sSCD)..." "...examine the prevalence of sSCD, its relation to objective cognitive change over time, and prospective associations with potential causative factors..." E.g. statistics: "...examined the prevalence of our main outcome measure, sSCD and its association with potential confounders. We examined the relationship between sSCD and objective cognitive decline... ...looked for an association between sSCD and cognitive internal inconsistency... ... identify possible causal contributors to sSCD... ... SCD predicted a decline in CAMCOG..." 4. Are the methods described sufficiently to allow the study to be repeated? Minor comments: Which software has been used? Can you please highlight or make it clearer which participants of the CaPS are included in your study? At phase 5 (your starting point?) how many had Dem/CIND/ noSCD/sSCD, cognitively healthy/ overlap? and how did that look at follow-up? Why did you decide on defining decline as 1sd from the mean rate in your population vs. established (?) cut-off values for meaningful change on the items you used? 5. Are research ethics (e.g. participant consent, ethics approval) addressed appropriately? Minor comments: Did participants give consent at each phase of the study? If, yes, when the participant was identified as having dementia/ CIND or cognitive impairment, how was informed consent achieved?
--

	What is the ethics approval of CaPS versus your specific study? 7. If statistics are used are they appropriate and described fully? Minor comment related to question 1 (Q1): it would be easier to judge if it was clearer how the objectives/ RQs where related to each statistical method. L. 180 you have not stated how you determined the prevalence or its associations with confounders? L. 181. You have not stated how you examined the relationship between sSCD and objective cognitive decline? General, and linked to results, I struggle to comprehend which participants are included in which analyses (Dem/CIND, sSCD, no cognitive decline (subjective or objective)) at what phases (3--7?) of the CaPS, can you please make this clearer? 9. Do the results address the research question or objective? See comments linked to Q1 and Q7. 10. Are they presented clearly? See comments linked to Q1, Q7 and Q9. Specific comments: Supplemental Table 1: Are the numbers where columns noSCD and SCD don't add up to the total column missing participants? E.g. CIND N =192 (noSCD n=79, SCD n=71, ? n =42 [if missing 21.8% of people with CIND did not answer SCD question?]) – could that possibly introduce some bias in your results?]. Are there statistical differences between the characteristics between these groups (noSCD/SCD)? Supplemental Table 2: Did 13.1 % of people at phase 5 not answer your main outcome measure (SCD), do these participants differ from those who did answer, possible bias? Table 1. line 222 "...mildly worse..." aren't they statistical significantly worse? Since significance is not maintained when PwD are excluded, could subjective decline in CAMGOG be a proxy of objective decline in this analyses? (l. 334.?) Line 228–231 not results, move to discussion? Table 2. Since it's not part of any RQ (?) is it possible to remove all no objective decline (and ve-) from this table to make the results clearer? Table 3. N=? Table 4. N=? risk factors from P2-4 and established sSCD at P5 (n=326?) Why have you chosen to display unadjusted vs. adjusted, what does the unadjusted add to your paper (see Q11)? Have you done a sensitivity analysis of complete vs missing in adjusted in analyses in Table 4. (from 1225? You only have 601? left, have you introduced bias?)? Line. 257–260 I struggle to follow which participants are included in which analyses, where are these numbers from, could they be incorporated in the flow-chart? 11. Are the discussion and conclusions justified by the results? The discussion could benefit from being more tightly connected to the results, not all results are discussed. E.g. RQ10 Table 1? Additionally, is CAMCOG sensitive enough to capture early changes in cognition.? Reference line 301 and 302? Line 289–298. "SCD-plus" vs sSCD. sSCD more sensitive over time (reference)? But you also argue that sSCD is a normal consequence of ageing? Patients in Memory/ tertiary clinics are referred there because of their memory difficulties, whilst in a pop-based study you have quite a different sample altogether, do you have any thoughts
--	--

	about that? How does your sample, other than cognition differ from prev studies in clinics (age, education, comorbidity, physical activity, physical function)? E.g. table 4. How do you interpret what “looses” association when including all variable in the fully adjusted model? How do you interpret that results are the same when you exclude PwD? Do you have any theories why well-known risk factors for dementia is not predictors in population (are they healthier, not high enough scores on your outcomes?) ? The other variables are also risk factors for dementia (sleep, mood)? The conclusion and implications are a continuation of discussion and could benefit from being rewritten.
--	--

VERSION 1 – AUTHOR RESPONSE

Reviewer 1 comments

Introduction:

1.1 Since this study only included men, it will be helpful to include a discussion of potential gender differences from the literature, and included the justification of why only men, but not women, were examined.

ONLY MEN WERE EXAMINED DUE TO THE PRE-EXISTING DESIGN OF THE STUDY WHICH WAS TO STUDY PREDICTORS OF CORONARY HEART DISEASE. AS RATES OF HEART DISEASE IN WOMEN ONLY CATCH UP WITH MALE RATES IN THEIR MID-70s, THE ORIGINAL PI (PROF. PETER ELWOOD) DECIDED NOT TO RECRUIT WOMEN-(THEY HAD BEEN INCLUDED IN THE PILOT STUDY FOR CAPS) WE HAVE ADDED THE FOLLOWING SENTENCES.

P6: “Women were excluded as the initial focus was on cardiovascular health and it was felt that too few events would be accrued in the early follow-up.”

P18: “The association between subjective and objective cognition may be different in women, in particular since women may have an extra mid-life peak of subjective cognitive concern.(14)”

1.2 Line 77-78: it is unclear why depression was suddenly brought up in the discussion without prior context. It would be helpful to provide the relationship of depression to this study.
THIS HAS BEEN RE-WORDED TO FOCUS MORE CLEARLY ON SUBJECTIVE AND OBJECTIVE COGNITION.

1.3 It is a bit unclear why SCD-plus was mentioned in the article since it is not the definition used for this study. The author could consider adding a summary of what message they were trying to bring in a sentence that follows the discussion.

SCD-PLUS WAS MENTIONED SINCE IT WOULD BE OVER-REPRESENTED IN THE CLINIC VERSUS THE POPULATION, AND THEREFORE IS IMPORTANT IN INTERPRETING CLINIC-BASED PRIOR STUDIES. WE HAVE INCLUDED A SENTENCE AT THE END OF THIS PARAGRAPH P5: “But it is also likely that “SCD plus” would be over-represented in clinic populations relative to the general population.”

1.4 It would be better if the hypotheses of the study followed the aims of the study. This will strengthen the coherence of the introduction and help the reader to interpret the two.
WE HAVE RE-NUMBERED THE HYPOTHESES SO THEY NOW CORRESPOND TO SECTIONS/QUESTIONS ADDRESSED CONSECUTIVELY IN THE RESULTS.

Methods:

1.5 It would be helpful if authors could describe the tested components of the CAMCOG. Since the focus of this study is to examine the association between subjective and objective cognition, it is

important to understand whether the cognitive components examined in the assessment approaches are similar.

ALL COMPONENTS OF THE CAMCOG WERE TESTED AND THE SUMMED SCORE (CAMCOG) WAS ANALYSED. WE HAVE REMOVED MENTION OF "COMPONENTS OF" IN LINE 139 TO MAKE THIS MORE EXPLICIT.

1.6 A section of power analysis for the study is recommended.

IT IS NOT ROUTINE TO INCLUDE A POWER ANALYSIS WHEN UNDERTAKING EXPLORATORY SECONDARY DATA-ANALYSIS USING AN EXISTING DATASET WHERE THE SAMPLE SIZE IS FIXED AND THERE IS NO STRONG A PRIORI ON THE EXPECTED STRENGTH OF ASSOCIATION TO BE OBSERVED. BUT PLEASE SEE POINT 1.8 REGARDING A COMMENT ON POWER.

Results:

1.7 It would be helpful for interpretation if the authors could include the numerical analysis results (e.g., coefficient and p-value) to some of the important findings discussed in this section. THESE HAS NOW BEEN ADDED IN (P12-14).

Discussion:

1.8 Since a strength of this study is the longitudinal design; it would be better if the authors can leverage it further to discuss causality with reference to the observed associations.

THANKS FOR HIGHLIGHTING THIS. WE'VE ADDED (OR MOVED) SENTENCES IN THE DISCUSSION: "This is the first large population-based cohort study of subjective experience of cognitive decline which can look at both predictors as well as future clinical outcomes with a wide range of potential confounders measured at younger ages and hence less prone to reverse causation. We found no prospective association between subjective cognitive decline and later objective cognitive decline (either in numerical score or severity category i.e., CIND or dementia). Naturally, a larger sample size would have given greater power to detect an effect. However, this negative finding suggests that subjective cognitive symptoms by themselves, are not substantially (on a population level) the manifestation of a neurodegenerative condition."

1.9 The clinical implication section should be strengthened. Discussion on actionable intervention practice and assessment protocol in relation to the study findings are suggested.

WE HAVE STRENGTHENED THIS SECTION, TO INCLUDE THE IMPORTANCE OF DIAGNOSING FCD, AND THE NEED TO CONSIDER WHO MIGHT BE ELIGIBLE FOR EARLY TREATMENT OF NEURODEGENERATIVE CONDITIONS.

1.10 Immediate recall and attentional processes are subcomponents of executive functioning. What is the possible relevance of executive functioning deficit to the findings of this study?

WE DO NOT FEEL THAT ENOUGH PEOPLE IN THIS POPULATION SAMPLE WOULD HAVE HAD SIGNIFICANT PURELY EXECUTIVE FUNCTIONING DEFICITS, TO ACCOUNT FOR A HIGH PROPORTION OF THE IMMEDIATE RECALL AND ATTENTION EFFECTS SEEN.

1.11 After reading the discussion, I think adding more clarification about cognitive inconsistency will help. Looking at Table 3, inconsistency was defined as DR>IR. Justifications behind this formula are recommended in the main text. And it is also recommended to discuss why was cognitive inconsistency restricted to only memory components.

WE HAVE CLARIFIED THE NATURE OF INTERNAL INCONSISTENCY IN THE INTRODUCTION, AND ITS OPERALISATION AS DR>IR IN THE METHODS. IT WAS NOT POSSIBLE IN THE DATA WE HAD TO EXAMINE INTERNAL INCONSISTENCY IN OTHER COGNITIVE DOMAINS.

Reviewer: 2

Dr. Karen Sverdrup , Nasjonalt senter for aldring og helse, Oslo universitetssykehus Ullevål

2. 1. Is the research question or study objective clearly defined?

Comments:

The manuscript could benefit from coherent language in aims (line [l.] 91-92), objectives (l. 111-113?), hypothesis (l. 114-118), statistics (l. 180-188, what RQs are being answered) and subheading of results (l. 206, 218, 235-236, 245, 256). Could the objectives be reframed to research questions (RQs) in line with results subheadings?

E.g. intro:

"...explore population-based associations with, and predictions from, the subjective experience of cognitive decline (Sscd)..." "...examine the prevalence of Sscd, its relation to objective cognitive change over time, and prospective associations with potential causative factors..."

E.g. statistics:

"...examined the prevalence of our main outcome measure, Sscd and its association with potential confounders. We examined the relationship between Sscd and objective cognitive decline... ...looked for an association between Sscd and cognitive internal inconsistency... ... identify possible causal contributors to Sscd... ... SCD predicted a decline in CAMCOG..."

WE HAVE REFRAMED OUR HYPOTHESES AS RESEARCH QUESTIONS, WHICH NOW CORRESPOND TO THE STATISTICAL ANALYSIS PLAN, AND SUBHEADINGS IN THE RESULTS. WE HAVE REVISED THE HIGHLIGHTED SECTIONS.

(NOTE THERE WERE NO QUERIES NUMBERED "2" OR "3" FROM THIS REVIEWER; SIMILARLY NONE FOR "6" & "8" BELOW):

2.4. Are the methods described sufficiently to allow the study to be repeated?

Minor comments:

Which software has been used?

STATA VERSION 16 – THIS IS NOW MENTIONED IN THE TEXT.

Can you please highlight or make it clearer which participants of the CaPS are included in your study? YES WE HAVE NOW CLARIFIED THIS IN THE STATISTICAL ANALYSIS PLAN (p9-10). THE STUDY FLOWCHART IS ALSO AVAILABLE AS SUPPLEMENTAL FIGURE 1.

At phase 5 (your starting point?) how many had Dem/CIND/ noSCD/Sscd, cognitively healthy/ overlap? And how did that look at follow-up?

THIS INFORMATION IS ALREADY INCLUDED IN THE FIRST TWO PARAGRAPHS OF THE RESULTS, AND SUPPLEMENTAL TABLE 1.

Why did you decide on defining decline as 1sd from the mean rate in your population vs. established (?) ut-off values for meaningful change on the items you used?

WE AGREE 1 SD IS RELATIVELY ARBITRARY BUT IS REGARDED AS LARGER VARIABILITY THAN MIGHT BE EXPECTED FROM SIMPLE RANDOM VARIABILITY ON IMMEDIATE TEST-RETEST. WE WERE LOOKING FOR POTENTIALLY SUBTLE CHANGES THAT MIGHT BE MISSED IN CLINICAL PRACTICE AS WE HAD THE BENEFIT OF DETAILED COGNITIVE ASSESSMENTS PREDATING COGNITIVE CONCERNS. WE HAVE CLARIFIED THIS ON p7-8.

2.5. Are research ethics (e.g. participant consent, ethics approval) addressed appropriately? Minor comments:

Did participants give consent at each phase of the study? If, yes, when the participant was identified as having dementia/ CIND or cognitive impairment, how was informed consent achieved?

What is the ethics approval of CaPS versus your specific study?

THIS SPECIFIC STUDY INVOLVED SECONDARY DATA ANALYSIS OF PREVIOUSLY COLLECTED DATA, AND WAS APPROVED BY THE CaPS STEERING COMMITTEE. PHASE V DATA COLLECTION WAS APPROVED BY THE GWENT RESEARCH ETHICS COMMITTEE. PARTICIPANTS WITH SEVERE COGNITIVE PROBLEMS DID NOT USUALLY PARTICIPATE IN THE FOLLOW-UP FOR OBVIOUS REASONS. ALL THE PARTICIPANTS WERE ASSESSED FOR CAPACITY TO CONSENT BY THE FIELDWORKER WHO UNDERTOOK THE COGNITIVE TESTING AND PROVIDED THEIR CONSENT FOR PARTICIPATION AT EACH PHASE.

2.7. If statistics are used are they appropriate and described fully?

Minor comment related to question 1 (Q1): it would be easier to judge if it was clearer how the objectives/ RQs were related to each statistical method.

L. 180 you have not stated how you determined the prevalence or its associations with confounders?

L. 181. You have not stated how you examined the relationship between Sscd and objective cognitive decline? General, and linked to results, I struggle to comprehend which participants are included in which analyses (Dem/CIND, Sscd, no cognitive decline (subjective or objective)) at what phases (3–7?) of the CaPS, can you please make this clearer?

THESE AREAS HAVE BEEN AMENDED TO MAKE THIS INFORMATION CLEAR, BOTH IN THE STATISTICAL ANALYSIS PLAN, AND IN RE-ORGANISING 4 RESEARCH QUESTIONS TO LINK UP TO 4 SECTIONS WITHIN THE RESULTS.

2.9. Do the results address the research question or objective?

See comments linked to Q1 and Q7.

2.10. Are they presented clearly?

See comments linked to Q1, Q7 and Q9. Specific comments:

Supplemental Table 1: Are the numbers where columns noSCD and SCD don't add up to the total column missing participants? E.g. CIND N =192 (noSCD n=79, SCD n=71, ? n =42 [if missing 21.8% of people with CIND did not answer SCD question?]) – could that possibly introduce some bias in your results?. Are there statistical differences between the characteristics between these groups (noSCD/SCD)? Supplemental Table 2: Did 13.1 % of people at phase 5 not answer your main outcome measure (SCD), do these participants differ from those who did answer, possible bias?

YOU ARE CORRECT, SOME PARTICIPANTS HAVE MISSING DATA FOR THE QUESTIONS ABOUT SCD (PHASE 5), THIS IS WHY SUPPLEMENTAL TABLE 1 DOES NOT ADD UP TO THE TOTAL. WE HAVE HIGHLIGHTED THIS UNDER THE TABLE, AND IN RESULTS (P11). SOME MISSING DATA WOULD BE INEVITABLE IN A PROSPECTIVE COHORT OF ELDERLY PEOPLE, SOME OF WHOM HAVE COGNITIVE PROBLEMS. WE THINK IT IS LIKELY THESE DATA ARE MISSING AT RANDOM (MAR). OUR ANALYSES THAT EXCLUDE THOSE WITH DEMENTIA PROVIDE SOME PROTECTION AGAINST THE POTENTIAL BIAS OF MISSINGNESS ON SCD BEING DUE TO SEVERE COGNITIVE IMPAIRMENT (WE HAVE EXPLAINED THIS ON P10). IN ADDITION, WE HAVE ADDED MULTIPLE IMPUTATION TO TABLE 4 (PLEASE SEE BELOW).

Table 1. line 222 "...mildly worse..." aren't they statistical significantly worse?

THIS WORDING HAS BEEN CHANGED TO "slightly worse".

Since significance is not maintained when PwD are excluded, could subjective decline in CAMGOG be a proxy of objective decline in this analyses? (l. 334.?)

THE INTERPRETATION OF THE ASSOCIATION BETWEEN SUBJECTIVE AND OBJECTIVE COGNITION IS COMPLEX, SINCE SEVERELY IMPAIRED PEOPLE MIGHT LACK INSIGHT OR ABILITY TO UNDERSTAND THE QUESTION, BUT ALSO THERE ARE A LARGE NUMBER OF

PEOPLE IN THE GENERAL POPULATION WHO DO FEEL THEIR COGNITION IS SUBOPTIMAL. AS ABOVE, OUR ANALYSES EXCLUDING THOSE WITH DEMENTIA PROVIDE SOME EVIDENCE TO TEST WHETHER THE RESULTS ARE DRIVEN BY A BIAS (MISSING DATA ON SCD BEING A POTENTIAL INDICATOR OF SEVERE OBJECTIVE COGNITIVE IMPAIRMENT). THE OVERALL MESSAGE FROM TABLES 1 AND 2 IS THAT WHILST THERE IS AN ASSOCIATION BETWEEN OBJECTIVE AND SUBJECTIVE COGNITION, IT IS MODEST.

Line 228–231 not results, move to discussion?

THIS HAS BEEN MOVED TO DISCUSSION AS SUGGESTED.

Table 2. Since it's not part of any RQ (?) is it possible to remove all no objective decline (and ve-) from this table to make the results clearer?

WE RESPECTFULLY DISAGREE AND THINK IT IS HELPFUL TO RETAIN THE "NO OBJECTIVE DECLINE" COLUMN IN ORDER TO DEMONSTRATE BOTH THE 'FALSE POSITIVES' AND 'FALSE NEGATIVES' THAT WOULD OCCUR FROM ASSUMING SUBJECTIVE REPORTS ARE A PROXY FOR OBJECTIVE DECLINE.

Table 3. N=?

APOLOGIES FOR THIS OMMISION, THIS HAS NOW BEEN ADDED IN (N=619).

Table 4. N=? risk factors from P2-4 and established Sscd at P5 (n=326?) Why have you chosen to display unadjusted vs. adjusted, what does the unadjusted add to your paper (see Q11)?

DISPLAYING THE UNADJUSTED AS WELL AS THE ADJUSTED MODEL DISPLAYS THE CRUDE RELATIONSHIPS BETWEEN VARIABLES. IT ALLOWS ONE TO EMPIRICALLY ASSESS THE SIZE AND DIRECTION OF CONFOUNDING WHICH IS SPECIFICALLY HELPFUL IN UNDERSTANDING THE IMPORTANCE OF ANY POTENTIAL RESIDUAL CONFOUDNING (E.G. IF AN ADJUSTED ODDS RATIO WAS 3.5 AND AFTER ADJUSTING FOR A CRUDE MEASURE OF SMOKING STATUS (SMOKER YES/NO) IT WAS ATTENUATED TO 1.7 (BUT REMAINED STATISTICALLY SIGNIFICANT) IT IS LIKELY THAT BETTER ADJUSTMENT USING MORE DETAILED CIAGRETTES PER DAY WOULD FURTHER ADJUST THE ASSOCIATION TO THE NULL. IF ON THE OTHER HAND ADJUSTMENT SHIFTED THE EFFECT ESTINMATE FROM 3.5 TO 3.3 THEN THERE IS LESS CONCERN ABIOUT RESIDUAL CONFOUNDING. WE WILL MOVE THIS TO THE SUPPLEMENTAL TABLE 3. THE N FOR EACH MODEL IS LISTED IN THE TABLE HEADER.

Have you done a sensitivity analysis of complete vs missing in adjusted in analyses in Table 4. (from 1225? You only have 601? Left, have you introduced bias?)?

THIS MODEL HAS NOW BEEN RE-RUN USING MULTIPLE IMPUTATION WITH CHAINED EQUATIONS (N=1,225). THE RESULT IS NOW IN SUPPLEMENTARY TABLE 3. THE VARIABLES THAT REMAIN AS SIGNIFICANT INDEPENDENT VARIABLES ARE THE SAME AS IN THE COMPLETE CASE ANALYSIS, EXCEPT WITH THE ADDITION OF "RATE OF DECLINE IN AH4". A COMMENT ON THIS HAS BEEN ADDED TO THE RESULTS SECTION (P14).

Line. 257–260 I struggle to follow which participants are included in which analyses, where are these numbers from, could they be incorporated in the flow-chart?

WE HAVE UPDATED THE RESULTS SUB-HEADINGS TO MAKE IT CLEAR WHICH PHASES THE DATA IS FROM. WE HAVE MOVED THE LINES MENTIONED HIGHER UP IN THE TEXT TO MAKE IT CLEARER WHAT IS THE ANSWER TO THE RESEARCH QUESTION.

WE HAVE REMOVED THE INTERACTION ANALYSIS FROM TABLE 5 AS THIS WAS A POST HOC ANALYSIS AND OVERALL WE FELT IT DID NOT ADD TO THE READER'S UNDERSTANDING OF THE MANUSCRIPT.

11. Are the discussion and conclusions justified by the results?

The discussion could benefit from being more tightly connected to the results, not all results are discussed.

E.g. RQ10 Table 1?

WE HOPE THIS IS NOW CLEARER SINCE WE HAVE RE-ORGANISED THE RESULTS SECTION (WHERE TABLES ARE HIGHLIGHTED) INTO SEGMENTS PER RESEARCH QUESTION.

Additionally, is CAMCOG sensitive enough to capture early changes in cognition.?

THAT IS AN EMPIRICAL QUESTION, AND WE HAVE EXPLORED THIS POSSIBILITY IN THE DISCUSSION (LINE 392-7).Reference line 301 and 302?

THIS INFORMATION IS CLEAR FROM THE REFERENCES TO THESE COGNITIVE ASSESSMENTS GIVEN IN THE METHODS SECTION, THESE REFERENCES ARE NOW REFERRED TO AGAIN IN THE DISCUSSION (RELEVANT LINES NOW 393-4).

Line 289–298. “SCD-plus” vs Sscd. Sscd more sensitive over time (reference)? But you also argue that Sscd is a normal consequence of ageing?

WE HAVE RE-WORDED THIS SENTENCE (385-8) TO HIGHLIGHT WE ARE NOT CLAIMING A DIRECT COMPARISON BETWEEN Sscd AND SCD-PLUS IN SENSITIVITY, RATHER HIGHLIGHTING THE POTENTIAL ADVANTAGES OF OUR APPROACH.

Patients in Memory/ tertiary clinics are referred there because of their memory difficulties, whilst in a pop-based study you have quite a different sample altogether, do you have any thoughts about that? How does your sample, other than cognition differ from prev studies in clinics (age, education, comorbidity, physical activity, physical function)?

THIS IS INDEED AN IMPORTANT ASPECT OF OUR STUDY, AND WE HAVE NOW INCLUDED A SENTENCE HIGHLIGHTING THESE POTENTIAL DIFFERENCES IN THE DISCUSSION (372-379).

E.g. table 4. How do you interpret what “looses” association when including all variable in the fully adjusted model? How do you interpret that results are the same when you exclude PwD?

WE HAVE UPDATED SENTENCES IN THE DISCUSSION TO HIGHLIGHT OUR INTERPRETATION (432-435): “However, factors typically associated with neurodegenerative change (vascular health, alcohol use) did not predict later sSCD in the fully adjusted model. This also held in an analysis excluding people with dementia. We interpret this to indicate that subjective cognitive decline is predicted by objective cognitive decline and its determinants, but also has important predictors independent of objective cognition.”

Do you have any theories why well-known risk factors for dementia is not predictors in population (are they healthier, not high enough scores on your outcomes?) ? The other variables are also risk factors for dementia (sleep, mood)?

PLEASE SEE REASONING ABOVE

VERSION 2 – REVIEW

REVIEWER	Stephen C L Lau Washington University School of Medicine in Saint Louis
REVIEW RETURNED	09-Aug-2023

GENERAL COMMENTS	Thank you for giving me the opportunity to review this revised manuscript. The authors have adequately addressed most of my concerns in the revision, except for the issue on power. The authors have declined to conduct a power analysis because of 1. the use of secondary data and 2. the exploratory nature of the study. However, I am afraid that these reasons are not convincing. Bierman (2007)
---

	recommended that power analysis is needed for secondary data in order to avoid Type II errors, just as with new investigations. At minimum, the authors should calculate the minimum sample sizes needed to perform such investigations or comment about this to help readers when interpreting the findings of the study.
--	--

VERSION 2 – AUTHOR RESPONSE

“Thank you for giving me the opportunity to review this revised manuscript. The authors have adequately addressed most of my concerns in the revision, except for the issue on power. The authors have declined to conduct a power analysis because of 1. the use of secondary data and 2. the exploratory nature of the study. However, I am afraid that these reasons are not convincing. Bierman (2007) recommended that power analysis is needed for secondary data in order to avoid Type II errors, just as with new investigations. At minimum, the authors should calculate the minimum sample sizes needed to perform such investigations or comment about this to help readers when interpreting the findings of the study.”

Thank you for this reference, unfortunately this appears to be to a book chapter not available to us to view. Hence we have not been able to cite it. However we have cited a 2020 paper that cites Bierman, discussing the issue of post hoc power calculations to avoid type II errors.

We have now presented post hoc power analyses, examining power to detect differential risk of incident CIND or dementia across those with or without sSCD, at differing levels of risk difference. We also highlighted the implications of this in the Limitations section.

VERSION 3 – REVIEW

REVIEWER	Stephen C L Lau Washington University School of Medicine in Saint Louis
REVIEW RETURNED	11-Sep-2023
GENERAL COMMENTS	Thank you to the authors who have adequately addressed all my comments.